# High proportion of RR-TB and mutations conferring RR outside of the RRDR of the rpoB gene detected in GeneXpert MTB/RIF assay positive pulmonary tuberculosis cases, in Addis Ababa, Ethiopia

**Gizachew Taddesse Akalu**[1,2]*, **Belay Tessema**[3], **Beyene Petros**[1]

1 Department of Microbial, Cellular, and Molecular Biology, Faculty of Life Sciences, College of Natural and Computational Sciences, Addis Ababa University, Addis Ababa, Ethiopia, 2 Department of Microbiology, Immunology, and Parasitology, St. Paul's Hospital Millennium Medical College, Addis Ababa, Ethiopia, 3 Department of Medical Microbiology, School of Biomedical and Laboratory Medicine, University of Gondar, Gondar, Ethiopia

* gtakalu@gmail.com

**Data Availability Statement:** All relevant data are within the paper and its Supporting Information files.

## Abstract

### Background

Globally, TB is the leading cause of infectious disease morbidity and mortality with many diagnostic uncertainties. Access to affordable and rapid diagnostics remained a major challenge for many developing countries which bear the greatest burden of TB delaying the initiation time to treatment.

### Objective

This study aimed to assess the GeneXpert MTBRIF assay probe utility for the detection of pulmonary TB and Rifampicin-resistant TB cases in Addis Ababa, Ethiopia.

### Materials and methods

A cross-sectional study was performed from October 2019 to July 2020 in Saint Peter TB Specialized Hospital in Addis Ababa metropolitan area, Ethiopia. This study enrolled 216 clinically suspected new presumptive pulmonary TB cases confirmed by GeneXpert MTB/RIF Assay. Sociodemographic and clinical characteristics were captured using a structured tool. Data were entered in Microsoft Excel 2019, checked for inconsistency, cleaned promptly, and exported to IBM SPSS Statistics for Windows, Version 26.0. Armonk, N.Y: IBM Corp, the USA for analysis. Descriptive analysis and binary and multivariate logistics regression were performed and all statistical significance was determined at a 95% confidence level.

### Results

The majority of the study participants, 55.1% [119/216] were males aged 6–80 years. The prevalence of RR MTB was 11.11% [24/216]. A higher proportion of RR TB was found in

**Funding:** The authors received no specific funding for this work.

**Competing interests:** The authors have declared that no competing interests exist.

female patients [54.2%, 13/24], in patients in the age group of 30–50 years [45.8%, 11/24], in married individuals [62.5%, 15/24], in persons whose residence is urban [79.2%, 19/24], in persons who had a previous history of TB symptoms [100%, 24/24], in persons who had a history of contact with active and LTBI [33.3%, 8/24], and in persons who had a history of HIV and IDUs [41.7%, 10/24]. Occupation (AOR 22.868, 95% CI 1.655–316.022, p = 0.019), history of previous PTB+ (AOR 4.222, 95% CI 1.020–17.47, p = 0.047), and history of HIV and IDUs (AOR 4.733, 95% CI 1.416–15.819, p = 0.012) were independent predictors associated with RR-TB emergence. The commonest mutation 62.5% [15/24] was found in probe E (codons 529–533) region. There was no mutation associated with probe A (codons 507–511), probe B (codons 511–518), and probe C (codons 518–523) regions, as well as no combination of missed probes, was revealed. However, 12.5% [3/24] of RR TB patients were found without unidentified missed probe types detected outside of the RRDR. The delta Ct max was >4.0 and the highest proportion of 35.6% [77/216] RR TB was detected in samples of medium DNA load.

## Conclusion

The proportion of RR-TB we observed in this study was high. Similarly, a higher proportion of RR TB was detected outside of the RRDR. Moreover, a significant number of the GeneXpert MTB/RIF Assay probes were identified as unhybridized and this critical observation would mean that most of the probes had no or minimal utility in this geographical region. This calls for further studies to uncover mutation in the rpoB gene conferring RR and reshape TB triage and definite diagnostic algorithm in Ethiopia.

## Introduction

Tuberculosis (TB) is an infectious disease caused by a group of Mycobacterium species, the *Mycobacterium tuberculosis* (*M. tuberculosis)* complex [1]. Globally, TB is the leading cause of infectious disease morbidity and mortality with many diagnostic uncertainties [2, 3]. The global burden of latent tuberculosis infection (LTBI) is estimated to be approximately one-fourth of the global population and it is still a major public health interest as it can progress to active tuberculosis [4–7]. While TB is preventable and curable, it is yet the 13th leading cause of death and the second leading infectious cause of death following Coronavirus disease 19 worldwide [8].

In 2020, an estimated 10 million people were infected with TB worldwide, 5.8 million people were newly diagnosed with TB, 1.5 million deaths were recorded, and over 4 million of whom went undiagnosed predominantly due to the globally reduced access to TB diagnostic services [9]. Access to diagnostics remains a major challenge to developing countries which bear the greatest burden of TB delaying the initiation of timely treatment. Most of these deaths could have been prevented via early laboratory diagnosis which will further facilitate the way to appropriate treatment provision [8].

While TB infects humans globally, the disease is prevalent in 30 high TB-burden countries, accounting for 86% of TB infections worldwide. According to the WHO global TB report in 2020, Ethiopia is among the 30 high TB burden countries with an estimated annual TB incidence rate of 132 per 100 000. The number of people infected with TB is 151 000, 9 900 HIV positive TB incidence, 19 000 HIV negative TB mortality, 2 500 HIV positive TB mortality,

591 individuals diagnosed with DRTB, and 7 cases of laboratory-confirmed pre-XDR TB or XDR TB [9]. However, while the proportion of bacteriologically confirmed new TB cases was only 62% in 2020, neither the proportion of newly diagnosed TB cases using WHO recommended rapid diagnostics nor the percentage of TB cases who were placed on WHO recommended shorter regimen of drug-resistant TB is known. Recent studies showed that such large gaps in the detection and treatment of RR TB can result in resistant strains circulating within populations [10, 11]. Particularly in high TB burden countries, transmission was found to be the predominant cause of the globally rising rates of RR TB and MDR TB [12, 13]. Not invariably, reduced access to rapid TB diagnosis and treatment has resulted in an increase in TB deaths and disease transmission. The microbiological detection of TB is critical as it allows people to be correctly diagnosed to initiate the most effective treatment regimen early.

Historically, the detection of DR TB has been troublesome that requires expertise, a better laboratory platform to cultivate the bacterium, standard biosafety requirements, and a long turnaround time to result. Nevertheless, the development of GeneXpert MTB/RIF Assay was a significant step forward in improving the diagnosis of TB as well as the detection of mutations conferring rifampicin resistance TB simultaneously [14]. Detection of rifampicin resistance by GeneXpert MTB/RIF Assay is based on the absence or delay of binding of probes that cover the 81 bp RR determining region measured by a significant difference in the PCR threshold cycle value [15, 16].

In 2010, WHO endorsed the GeneXpert MTB/RIF Assay technology for detecting MTB and rifampicin resistance to facilitate rapid diagnosis of tuberculosis and early recognition of DR TB [17, 18]. Since then, the End TB Strategy puts renewed emphasis on the need to ensure early and correct diagnosis for all people with tuberculosis. Ethiopia has been implementing the WHO recommendations since 2012 and intensified when the Ethiopian National TB Control Program recommended that GeneXpert MTB/RIF Assay be used for testing on specimens from all presumptive TB patients irrespective of risk for DR-TB, HIV status, and age of the patient in situations where the GeneXpert MTB/RIF Assay System is accessible [19]. However, in Ethiopia, scientific evidence concerning the probes' utility and molecular epidemiology of RR TB evolving outside of the Rifampicin resistance determining region (RRDR) was not readily available to improve patient outcomes as well as TB surveillance in Ethiopia. This study sought to determine the GeneXpert MTB RIF assay probe utility for the detection of pulmonary TB and Rifampicin-resistant TB cases in the Addis Ababa, Ethiopia.

## Materials and methods

### Study design, setting, and study period

The study was performed from October 2019 to July 2020 in Saint Peter TB Specialized Hospital in Addis Ababa and adjacent town health centers in Sululta and Sendafa, Ethiopia. This was a prospective cross-sectional study enrolling 216 clinically suspected new presumptive TB patients. Saint Peter TB Specialized Hospital is a national reference center for TB diagnosis and treatment, providing care for seriously ill TB patients, and complications related to treatment. The hospital also serves as a referral TB center with a vision to be a center of excellence for TB diagnosis and treatment in East Africa. GeneXpert MTB/RIF Assay was implemented at the hospital in 2012 and has replaced smear microscopy as the primary diagnostic tool for an individual with clinical suspicion of presumptive TB.

### Study participants

The sources of the study population were all patients who visited Saint Peter TB Specialized Hospital in the Addis Ababa Metropolitan area, Ethiopia. Whereas, the study population was

all clinically suspected new pulmonary TB patients whose clinical suspicion was confirmed with GeneXpert MTB/RIF assay and referred to a TB clinic during the study period. The GeneXpert MTB RR results were the outcome or dependent variables of this study, whereas the sex, age, marital status, occupation, residence, religion, previous history related to TB symptoms, and behavior-related risk factors including perceived HIV status, intravenous drug use, tobacco smoking, alcohol use, and khat use were the independent variables or determinant associated factors which were investigated in this study.

## Inclusion and exclusion criteria

The inclusion criteria were clinical suspicion of new presumptive pulmonary TB strong enough to warrant a direct sputum investigation and a positive result confirmed by Gene Xpert MTM/RIF assay with informed consent. The exclusion criteria were being on treatment for TB within the last three months, and those who were unable to provide consent have been excluded.

## Sample collection

Each consenting, clinically suspected new presumptive TB patient was required to provide two sputum samples of 3-5ml each. The first sputum samples were collected upon enrollment, while the second sputum samples were collected in the morning the following day. Both sputum samples were expectorated and without any inducement or mechanical maneuver. Further, sociodemographic, and clinicoepidemiological data were captured using a structured questionnaire prepared for the purpose of this study.

## GeneXpert MTB/RIF assay

The GeneXpert MTB/RIF Assay analysis was performed in accordance with the protocol provided by the manufacturer; Cepheid, Sunnyvale, CA, USA [20]. Briefly, sputum samples were subjected to processing by approximately mixing with 8ml of GeneXpert sample reagent (SR) containing NaOH [5–10%] and isopropyl alcohol [10–20%] into the sputum container in a ratio of 2:1 V.V. The mixture was vigorously mixed by shaking the tubes 20 times and incubated for 15 minutes at 20–30˚C. This step was repeated in between the incubation time. The purpose of sputum processing was to facilitate digestion and liquefaction of organic debris that may be found in the specimen and decontaminating bacteria other than mycobacteria. Following sample preparation, approximately 3ml of the liquified specimen was transferred into the GeneXpert MTB/RIF cartridge with proper labeling and loaded into the GeneXpert Instrument System. By starting the test on the system software, the GeneXpert Machine automates all steps, including sample workup, nucleic acid amplification, detection of the target sequence, and result interpretation.

## GeneXpert MTB/RIF assay performance characteristics and result interpretation

The test result indicates the presence or absence of *M. tuberculosis* complex DNA with a qualitative and semiquantitative estimate of bacillary load and the presence or absence of the most common mutations causing rifampicin resistance. Generally, results from GeneXpert MTB/RIF Assay were obtained and assessed with three result types: MTB detected RR detected, MTB detected RR not detected and Invalid/Error/No result. In addition, four semiquantitative categories of results were obtained and analyzed when the test detects MTB among probe types, DNA amount, and cycle thresholds values i.e., High [Ct value: <16], Medium [Ct value:

16–22], Low [Ct value: 22–28] and Very Low [Ct value: > 28]. Furthermore, we intentionally classified the observed results of mutations conferencing RR into missed probes and without missed probes or unidentified probes to illicit insights for a better understanding of the susceptibility and probes' utility.

## Data analysis

Data were entered in Microsoft Excel 2019, checked for inconsistency, and cleaned promptly. We exported it to IBM SPSS Statistics for Windows, Version 26.0. Armonk, N.Y: IBM Corp, the USA for analysis. Descriptive parameters such as relative frequencies and percentages were used to describe patient characteristics, set the number and the proportion of cases, and describe qualitative variables. Whilst mean and standard deviation was used to describe continuous variables. Binary logistic regression was performed to identify the associated factors of bacteriologically confirmed MTB and RR MTB results. A multivariate logistics regression model was computed by considering associated factors found to be significant at P-value <0.05 with a Chi-square test to identify independent predictors for RR MTB detection. All statistical significance was determined at a 95% confidence level.

## Quality assurance

To monitor the quality of the samples and to maintain a high standard of accuracy of results, standard operating procedures were strictly followed in sputum collection, processing, and during laboratory analysis. The internal quality control of the GeneXpert MTB/RIF Assay System was validated using *M. tuberculosis* H37Rv ATCC laboratory reference strain, a non-RIF resistant and known RIF resistant strain stored at -20°C when a new batch of cartilages started. Moreover, GeneXpert MTB/RIF Assay System has inbuilt internal quality monitoring systems: Sample Processing Control and a Probe Check Control both usually included in the cartridge. The SPC is present to control for adequate processing of the target bacteria and to monitor the presence of inhibitors in the PCR reaction. Whereas, the PCC verifies reagent rehydration, PCR tube filling in the cartridge, probe integrity, and dye stability. Nevertheless, all specimens suspected of containing *M. tuberculosis* were handled with appropriate precaution at all times and manipulated only within an appropriate biosafety cabinet.

## Ethical considerations

The study was approved by the Institutional Review Board of the Department of Microbial, Cellular, and Molecular Biology, Faculty of Life Sciences, College of Natural and Computational Sciences, Addis Ababa University with Protocol No. IRB/039/2019, in April 04, 2019. In addition, a permission letter was obtained from Saint Peter TB Specialized Hospital before the commencement of the study. All study participants provided written informed consent. All subject identifiers were delinked from the source file and confidentiality and anonymity were ensured throughout the data analysis and study period.

## Results

### Sociodemographic characteristics of study participants

Overall, a total of 216 participants who had enrolled in this study were considered persons with presumptive TB upon clinical suspicion and bacteriologically confirmed with GeneXpert MTB RIF Assay. The age of the study subjects ranges from 6–80 years, with a mean and standard deviation of 34.65 and 13.885 years respectively. Among all the study participants, males account for 55.1% [119/216] with a ratio of 1.22:0.82. The average income of the study

participants ranges from 0 to 9,500 Ethiopian Birr (ETB) per month with a mean and standard deviation of 1175.00 and 1621.089, respectively (Table 1).

## Prevalence of RR TB detected in relation to known risk factors

Among the clinically suspected and bacteriologically confirmed MTB positives cases, the prevalence of MTB-detected Rifampicin resistance detected (MTB/RR TB) was found in 11.11% [24/216]. The majority, 54.2% [13/24] of RR TB was found from SPH, 54.2% [13/24] in female patients, 45.8% [11/24] in patients with an age group of 30–50 years old, 62.5% [15/24] in married individuals, 79.2% [19/24] in persons whose residence is urban, 70.8% [17/24] in Christians by religion, and 54.2% [13/24] in those with < = 500 ETB monthly income. Further, MTB/RR TB was detected in 100% [24/24] of persons who had a previous history of TB symptoms, 33.3% [8/24] in persons who had a history of contact with active TB patients as well as treatment history for active and LTBI, 66.7% [16/24] in individuals who had a history of chest X-ray for PTB, 41.7% [10/24] in persons who reported history of HIV and intravenous drug users, 20.8% [5/24] in persons who had a previous history of hospital admission as well as Khat use and 12.5% [3/24] of persons who had a previous history of smoking. Occupation (AOR 22.868, 95% CI 1.655–316.022, p = 0.019), history of previous PTB+ (AOR 4.222, 95% CI 1.020–17.47, p = 0.047), and history of HIV and IDUs (AOR 4.733, 95% CI 1.416–15.819, p = 0.012) were independent predictors associated with RR TB emergence (Table 2).

## The proportion of RR-TB among missed probes types and RR-TB evolving to escape from RRDR of 81 bp rpoB gene

The GeneXpert MTB/RIF assay utilizes molecular beacon technology to detect DNA sequences amplified in a hemi-nested RT-PCR assay. Usually, five different nucleic acid hybridization probes labeled as A, B, C, D, and E are used in the same multiplex reaction in which each probe is complementary to a different target sequence within the rpoB gene of rifampicin susceptible *M. tuberculosis*. Altogether, these overlapping probes span the entire 81 bp core region of the rpoB gene. Whilst dealing with the frequencies of different probe mutations, the commonest mutation that confers RR TB accounted for 62.5% [15/24] and was found in probe E (codons 529–533) region, followed by probe D (codons 523–529) region which accounted for 25.0% [6/24]. However, there was no mutation observed associated with probe A (codons 507–511), probe B (codons 511–518), and probe C (codons 518–523) regions, as well as no combination of multiple missed probe type was observed. Nonetheless, 12.5% [3/24] of mutations conferring RR TB were found without unidentified missed probe types and this could be due to the possibility of the existence of rpoB mutations outside RRDR or a different mechanism of rifampicin resistance in this setting (Fig 1 and Table 3).

## DNA amount and proportion of cycle threshold value across the study participants

The basis for the detection of rifampicin resistance is the difference between the first (early CT) and the last (late CT) specific *M. tuberculosis* beacon (ΔCT). The valid maximum cycle threshold (Ct) of 39.0 for Probes A, B, and C and 36.0 for Probes D and E were set for MTB/RIF data analysis. In addition, the delta Ct max was >4.0. The GeneXpert MTB/RIF assay provides the amount of DNA semi-quantitatively. Accordingly, the proportion of RR TB detected with high, medium, low, and very low DNA amounts were 22.2% [48/216], 35.6% [77/216], 23.6% [51/216], and 18.5% [40/216] respectively (Fig 2).

**Table 1. Sociodemographic characteristics and proportion of MTB/RR TB among study participants in Addis Ababa from October 2019 to March 2020.**

| Variables | Classification | Total Frequency | MTB/RR MTB | | | |
| --- | --- | --- | --- | --- | --- | --- |
| | | | Detected | | Not Detected | |
| | | | Number | Percent | Number | Percent |
| **Sampling Site** | SPH[a] | 131 [60.6] | 13 | 6.0 | 118 | 54.6 |
| | SuHC[b] | 39 [18.1] | 4 | 1.9 | 35 | 16.2 |
| | SeHC[c] | 46 [21.3] | 7 | 3.2 | 39 | 18.1 |
| **Sex** | Male | 119 [55.1] | 11 | 5.1 | 108 | 50.0 |
| | Female | 97 [44.9] | 13 | 6.0 | 84 | 38.9 |
| Age Group | < = 29 Years | 99 [45.8] | 10 | 4.6 | 89 | 41.2 |
| | 30–50 Years | 86 [39.8] | 11 | 5.1 | 75 | 34.7 |
| | > = 51 Years | 31 [14.4] | 3 | 1.4 | 28 | 13.0 |
| Marital Status | Single | 79 [36.6] | 9 | 4.2 | 70 | 32.4 |
| | Married | 130 [60.1] | 15 | 6.9 | 115 | 53.2 |
| | Divorced | 7 [3.2] | 0 | 0.0 | 7 | 3.2 |
| Residence | Urban | 161 [74.5] | 19 | 8.8 | 142 | 65.7 |
| | Rural | 55 [25.4] | 5 | 2.3 | 50 | 23.1 |
| Religion | Christian | 183 [84.7] | 17 | 7.9 | 166 | 76.8 |
| | Muslim | 33 [15.2] | 7 | 3.2 | 26 | 12.0 |
| Occupation | House Wife | 33 [15.3] | 5 | 2.3 | 28 | 13.0 |
| | Daily Laborer | 77 [35.6] | 8 | 3.7 | 69 | 31.9 |
| | Employed | 68 [31.5] | 9 | 4.2 | 59 | 27.3 |
| | Unemployed | 38 [17.6] | 2 | 0.9 | 36 | 16.7 |
| Monthly Income in ETB | < = 500 | 110 [50.9] | 13 | 6.0 | 97 | 44.9 |
| | 501–1999 | 48 [22.2] | 6 | 2.8 | 42 | 19.4 |
| | 2000–10000 | 58 [26.8] | 5 | 2.3 | 53 | 24.5 |
| Previous History of TB Symptom | Yes | 213 [98.6] | 24 | 11.1 | 189 | 87.5 |
| | No | 3 [1.4] | 0 | 0.0 | 3 | 1.4 |
| History of Contact with Active TB Cases | Yes | 64 [29.6] | 8 | 3.7 | 56 | 25.9 |
| | No | 152 [70.4] | 16 | 7.4 | 136 | 63.0 |
| History of Any Medicine | Yes | 56 [25.9] | 9 | 4.2 | 47 | 21.7 |
| | No | 160 [74.0] | 15 | 6.9 | 145 | 67.1 |
| History of BCG Vaccination | Yes | 31 [14.3] | 5 | 2.3 | 26 | 12.0 |
| | No | 185 [85.6] | 19 | 8.8 | 166 | 76.8 |
| History of TB Skin Test | Yes | 23 [10.6] | 2 | 0.9 | 21 | 9.7 |
| | No | 193 [89.4] | 22 | 10.2 | 171 | 79.2 |
| History of TB IGRA Test | Yes | 29 [13.4] | 4 | 1.8 | 25 | 11.6 |
| | No | 187 [86.5] | 20 | 9.2 | 167 | 77.3 |
| History of Previous PTB+ | Yes | 36 [16.7] | 8 | 3.7 | 28 | 13.0 |
| | No | 180 [83.3] | 16 | 7.4 | 164 | 75.9 |
| Treatment History for Active and LTBI | Yes | 49 [22.7] | 8 | 3.7 | 41 | 19.0 |
| | No | 167 [77.3] | 16 | 7.4 | 151 | 69.9 |
| History of CXR for PTB | Yes | 133 [61.6] | 16 | 7.4 | 117 | 54.2 |
| | No | 83 [38.4] | 8 | 3.7 | 75 | 34.7 |
| History of Known Chronic Diseases | Yes | 42 [19.4] | 4 | 1.8 | 38 | 17.6 |
| | No | 174 [80.5] | 20 | 9.2 | 154 | 71.3 |
| Previous Rx and Dx History of CA | Yes | 10 [4.6] | 1 | 0.4 | 9 | 4.2 |
| | No | 206 [95.3] | 23 | 10.6 | 183 | 84.7 |

(*Continued*)

**Table 1.** (Continued)

| Variables | Classification | Total Frequency | MTB/RR MTB | | | |
| --- | --- | --- | --- | --- | --- | --- |
| | | | Detected | | Not Detected | |
| | | | Number | Percent | Number | Percent |
| History of HIV and IDUs | Yes | 49 [22.6] | 10 | 4.6 | 39 | 18.0 |
| | No | 167 [77.3] | 14 | 6.5 | 153 | 70.8 |
| History of Hospital Admission | Yes | 37 [17.1] | 5 | 2.3 | 32 | 14.8 |
| | No | 179 [82.9] | 19 | 8.8 | 160 | 74.1 |
| History of Alcohol Consumption | Yes | 54 [25.0] | 4 | 1.9 | 50 | 23.1 |
| | No | 162 [75.0] | 20 | 9.3 | 142 | 65.7 |
| History of Previous Khat Use | Yes | 44 [20.4] | 5 | 2.3 | 39 | 18.1 |
| | No | 172 [79.6] | 19 | 8.8 | 153 | 70.8 |
| History of Smoking | Yes | 33 [15.3] | 3 | 1.4 | 30 | 13.9 |
| | No | 183 [84.7] | 21 | 9.7 | 162 | 75.0 |

[a]SPH: Saint Peter Hospital

[b]SuHC: Sululta Health Center

[c]SeHC: Sendafa Health Center

## Discussion

This study identified relevant evidence on sociodemographic characteristics, clinical features, and epidemiology of MTB and RR TB detected using GeneXpert which is a fully automated nested real-time PCR Assay. As per the WHO recommended rapid diagnostics for TB, and an updated national TB control program guideline in Ethiopia, GeneXpert MTB/RIF Assay has been endorsed for TB rapid diagnostics as well as prediction of RR TB from all presumptive TB patients irrespective of risk for DR TB, HIV status and age of the patient in situations where the instrument is accessible for better utility and patient care [17, 19].

Historically, drug resistance in *Mycobacterium tuberculosis* complex to a single treatment has emerged in different parts of the globe in the 1980s, and shortly after a few years then, the emergence of MDR TB become a global agenda known to be resistant to isoniazid and rifampicin which are the two most effective first-line anti-TB drugs. Since the 2000s XDR TB came into the global picture with bacteriological features of resistance not only to isoniazid and rifampicin but also to at least one fluoroquinolone and to one additional group A drug [21]. However, in the fight against TB, Rifampicin remains the most effective drug to treat the disease due to its bactericidal effects and it is the most valuable reserved drug for many countries. Nevertheless, recent studies showed that the rate of Rifampicin resistance is increasing due to missense mutations in a distinct 81 bp rpoB gene and an estimated 96% of all mutations are found in this RRDR region [22].

Early diagnosis and initiation of TB treatment are critical to reducing TB transmission, the incidence of infection, and the disease burden. The recently introduced rapid molecular TB diagnostics offer great promise for the TB control program [23]. However, Rifampicin resistance has mainly been associated with mutations in a limited region of the rpoB gene [24]. Many epidemiological surveys have been conducted in previous years to understand the nature and magnitude of MDR TB. However, only little is known regarding mutations conferring resistance to Rifampicin and the proportion of RR TB detected outside of the RRDR in Ethiopia. In this study, the prevalence of RR TB detected among bacteriologically confirmed new presumptive pulmonary TB patients using a rapid WHO-recommended GeneXpert MTB/RIF Assay was 11.1%. This result is comparably in line with previous studies in different

**Table 2. Factors associated with the outcome variables of RR TB, MTBC patients in Addis Ababa from October 2019 to March 2020.**

| Variables | Category | n [%] | COR at 95%CI | AOR at 95% CI | P. Value |
|---|---|---|---|---|---|
| | | | Binary Logistic Regression | Multinomial Logistic Regression | |
| Sampling Site | SPH[a] | 131[60.6] | 0.794 [0.484–1.304] | 0.350 [0.081–1.511] | 0.984 |
| | SuHC[b] | 39 [18.1] | | | |
| | SeHC[c] | 46 [21.3] | | | |
| Sex | Male | 119 [55.1] | 1.519 [0.648–3.562] | 0.828 [0.231–2.966] | 0.772 |
| | Female | 97 [44.9] | | | |
| Age Group | < = 29 Years | 99 [45.8] | 0.950 [0.524–1.721] | 1.907 [0.287–12.665] | 0.504 |
| | 30–50 Years | 86 [39.8] | | | |
| | > = 51 Years | 31[14.4] | | | |
| Marital Status | Single | 79 [36.6] | 1.177 [0.533–2.599] | 1 | 0.993 |
| | Married | 130 [60.2] | | | |
| | Divorced | 7 [3.2] | | | |
| Residence | Urban | 161[74.5] | 0.747 [0.265–2.107] | 2.618 [0.513–13.366] | 0.247 |
| | Rural | 55 [25.5] | | | |
| Religion | Christian | 183 [84.7] | 2.629 [0.994–6.952] | 0.316 [0.085–1.176] | 0.086 |
| | Muslim | 33 [15.3] | | | |
| Occupation | House Wife | 33 [15.3] | 1.254 [0.798–1.971] | 4.501[0.434–46.703] | 0.208 |
| | Daily Laborer | 77 [35.6] | | 3.903[0.475–32.102] | 0.205 |
| | Employed | 68 [31.5] | | 22.868[1.655–316.022] | **0.019** |
| | Unemployed | 38 [17.6] | | 1 | |
| Monthly Income in ETB | < = 500.00 | 110 [50.9] | 1.160 [0.693–1.941] | 2.250 [0.404–12.518] | 0.355 |
| | 501–1,999 | 48 [22.2] | | | |
| | 2,000–10,000 | 58 [26.9] | | | |
| Previous History of TB Symptom | Yes | 213 [98.6] | 0.000 | 0.000 | |
| | No | 3 [1.4] | | | |
| History of Contact with People with Active TB | Yes | 64 [29.6] | 0.824 [0.334–2.034] | 1.072 [0.317–3.624] | 0.911 |
| | No | 152 [70.4] | | | |
| History of Any Medicine | Yes | 56 [25.9] | 0.540 [0.222–1.315] | 1.381 0.448–4.259] | 0.575 |
| | No | 160 [74.1] | | | |
| History of BCG Vaccination | Yes | 31 [14.4] | 0.595 [0.204–1.732] | 1.346 [0.280–6.461] | 0.711 |
| | No | 185 [85.6] | | | |
| History of TB Skin Test | Yes | 23 [10.6] | 1.351 [0.296–6.157] | 0.427 [0.057–3.212] | 0.408 |
| | No | 193 [89.4] | | | |
| History of TB IGRA Test | Yes | 29 [13.4] | 0.749 [0.236–2.371] | 0.485 [0.089–2.656] | 0.404 |
| | No | 187 [86.6] | | | |
| History of Previous PTB+ | Yes | 36 [16.7] | 0.341 [0.134–0.873] | 4.222 [1.020–17.47] | **0.047** |
| | No | 180 [83.3] | | | |
| Treatment History for Active and LTBI | Yes | 49 [22.7] | 0.543 [0.217–1.357] | 1.311 [0.367–4.684] | 0.677 |
| | No | 167 [77.3] | | | |
| History of CXR TB | Yes | 133 [61.6] | 0.780 [0.318–1.912] | 1.185 [0.399–3.520] | 0.760 |
| | No | 83 [38.4] | | | |
| History of Chronic Diseases | Yes | 42 [19.4] | 1.234 [0.398–3.822] | 0.590 [0.143–2.437] | 0.466 |
| | No | 174 [80.6] | | | |
| Previous RX DX history for CA | Yes | 10 [4.6] | 1.131 [0.137–9.339] | 0.234 [0.013–4.228] | 0.325 |
| | No | 206 [95.4] | | | |
| History of HIV and IDUs | Yes | 49 [22.7] | 0.357 [0.147–0.864] | 4.733 [1.416–15.819] | **0.012** |
| | No | 167 [77.3] | | | |

(*Continued*)

**Table 2.** (Continued)

| Variables | Category | n [%] | COR at 95%CI | AOR at 95% CI | P. Value |
|---|---|---|---|---|---|
| | | | Binary Logistic Regression | Multinomial Logistic Regression | |
| History of Hospital Admission | Yes | 37 [17.1] | 0.760 [0.264–2.184] | 1.030 [0.198–5.375] | 0.972 |
| | No | 179 [82.9] | | | |
| History of Alcohol Consumption | Yes | 54 [25.0] | 1.761 [0.574–5.400] | 0.869 [0.128–5.909] | 0.886 |
| | No | 162 [75.0] | | | |
| History of Previous Khat Use | Yes | 44 [20.4] | 0.969 [0.340–2.757] | 0.703 [0.100–4.926] | 0.723 |
| | No | 172 [79.6] | | | |
| History of Smoking | Yes | 33 [15.3] | 1.296 [0.364–4.620] | 0.754 [0.081–6.987] | 0.803 |
| | No | 183 [84.7] | | | |

[a]SPH: Saint Peter Hospital

[b]SuHC: Sululta Health Center

[c]SeHC: Sendafa Health Center

subregions of Ethiopia reported by Mulu W, et al., (10.3%, 2017) [25], Derbie A, et al., (9.3%, 2016) [26], Wasihun AG, et al., (9.0%, 2020) [27], and Selfegna S, et al., (15.8%, 2022) [28]. A similar comparable proportion of RR TB has been reported in previous studies done in different high TB burden countries which include India by Kaur R, et al., (9.9%, 2016) [29] and

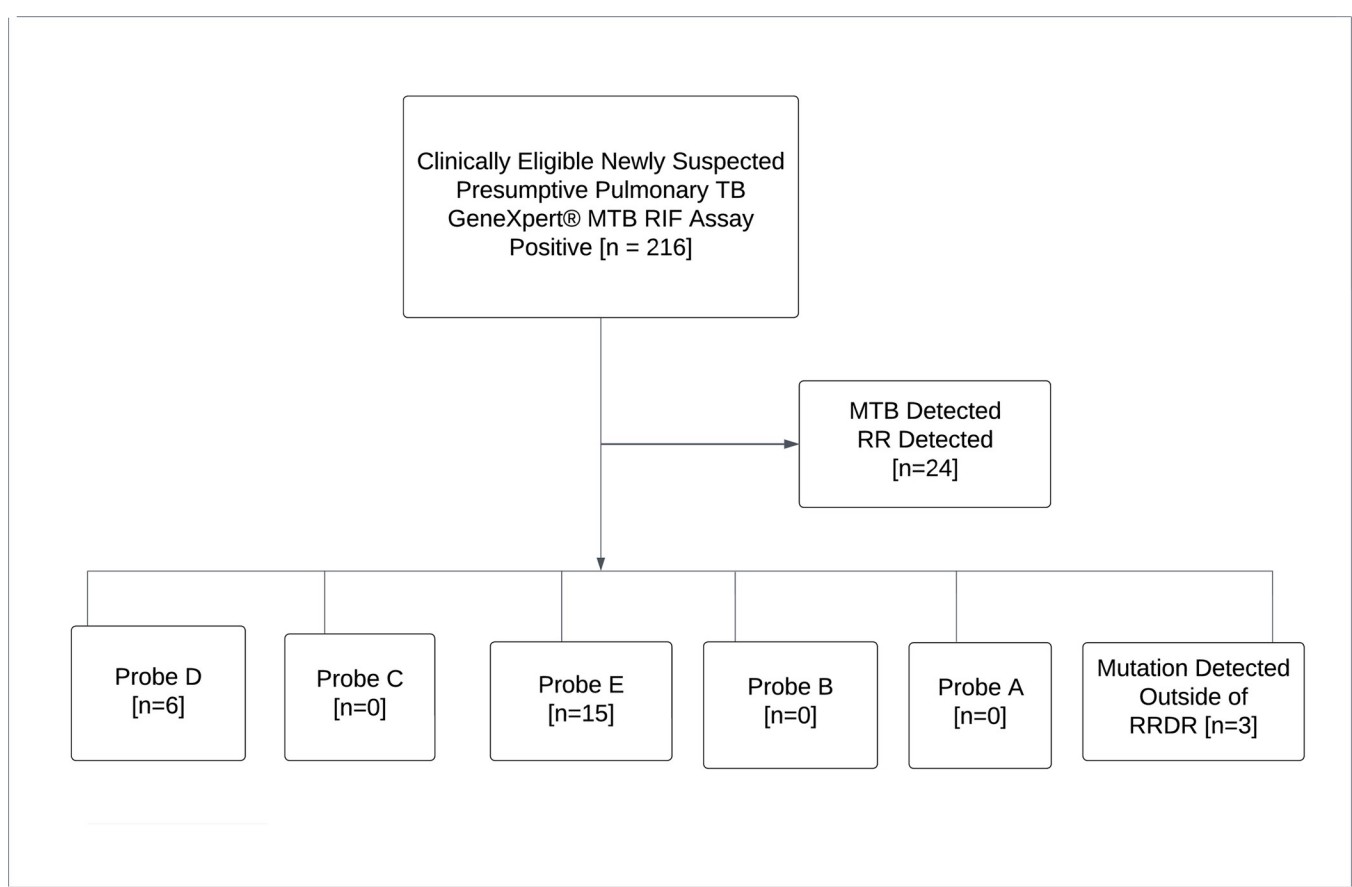

**Fig 1. Flowchart showing RR TB detected in GeneXpert MTB/RIF assay.**

**Table 3. The proportion of mutations conferring RR-TB detected in the RRDR in Addis Ababa from October 2019 to March 2020.**

| Probe Type | GeneXpert RR TB Mutation Analysis, n = 24 | | | |
| --- | --- | --- | --- | --- |
| | Missed Probe Result | | QC1, QC2 | Probe Check |
| | Negative | Proportion | | |
| Probe D [Codons 523–529] | 6 | 25 | NEG | Pass |
| Probe C [Codons 518–523] | . | . | NEG | Pass |
| Probe E [Codons 529–533] | 15 | 62.5 | NEG | Pass |
| Probe B [Codons 511–518] | . | . | NEG | Pass |
| Probe A [Codons 507–511] | . | . | NEG | Pass |
| RR without Missed Probes | 3 | 12.5 | NEG | Pass |
| Total | 24 | 100 | | |

Gupta A, et al., (10.5%, 2011) [30], in Nepal by Sah SK, et al., (10.2%, 2020) [31] and Joshi B, et al., (8.0%, 2018) [32], and in Nigeria by Nwadioha S, et al., (13.9%, 2014) [33] and Ikuabe PO, et al., (14.7%, 2018) [34].

However, this result is lower compared to results of the previous studies in other parts of Ethiopian subregions reported by Jaleta KN, et al., (15.8%, 2017) [35], Nigus D, et al., (18.2%,

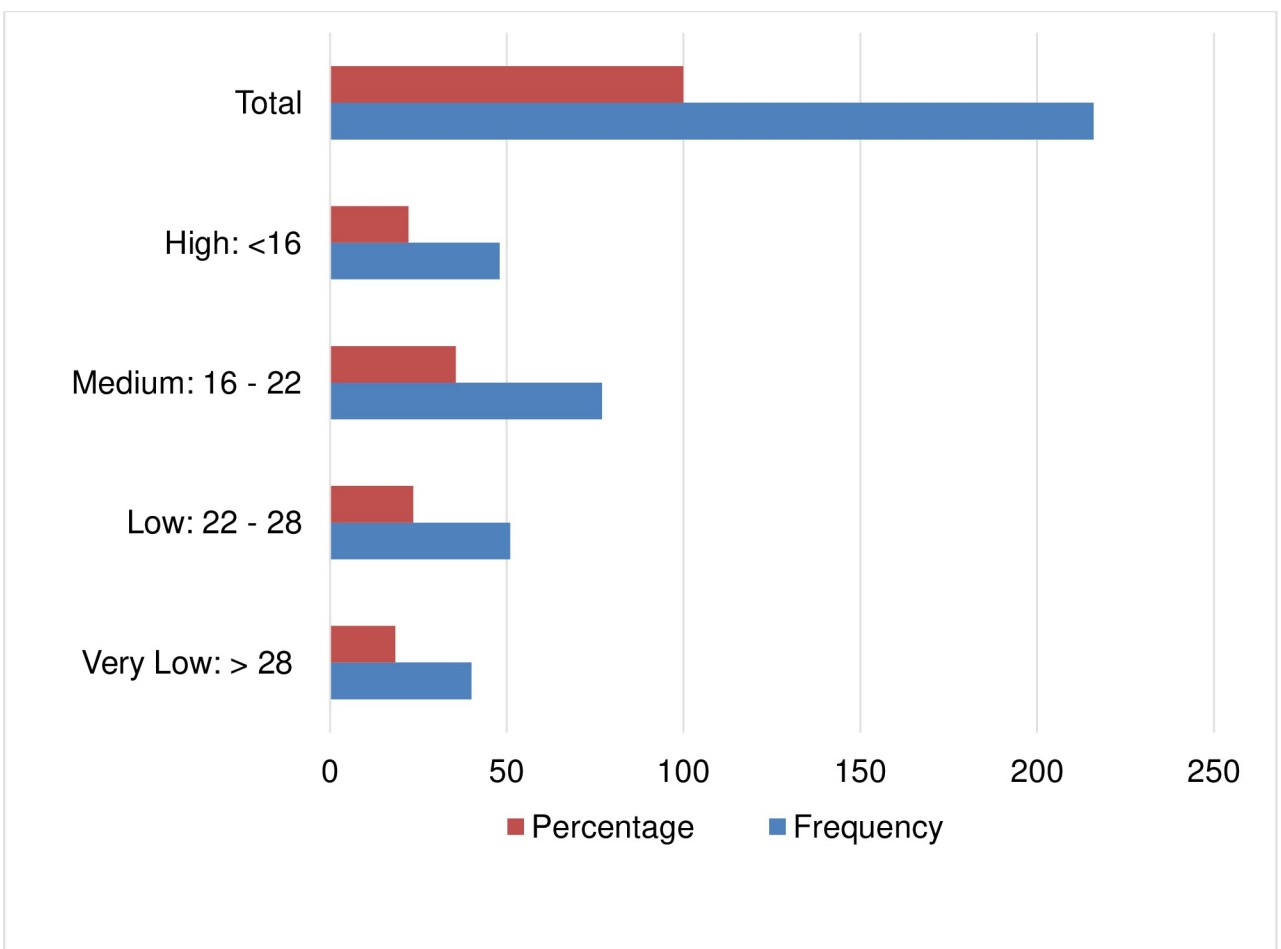

**Fig 2. DNA amount and cycle threshold value across the study subjects.**

2014) [36] and Mulisa G, et al., (33.2%, 2015) [37]. Similarly, our findings demonstrated a lower proportion of RR TB compared with previous study reports in Nigeria by Adejumo OA, et al., (23.4%, 2018) [38], in Congo by Farra A, et al., (42.2%, 2019) [39], and in Togo by Dagnra A, et al., (24%, 2015) [40].

Nevertheless, compared to other previous studies finding, the present study invariably showed a higher proportion of RR TB reported in other parts of Ethiopian subregions by Kuma D, et al., (5.1%, 2021) [41], and Admassu W, et al., (3.4%, 2022) [42]. Similarly, such reports in Uganda by Sekadde MP, et al., (5.7%, 2013) [43] and in Indonesia by Christopher PM, et al., (4.9%, 2019) [44] revealed a lower proportion of RR TB compared to our findings.

Contrary to the existing national as well as the global WHO estimated proportion of RR TB in new presumptive TB patients, the present study demonstrated a significantly higher proportion. Thus, the higher proportion of RR TB in this study would warrant finding a plausible diagnostic algorithm in order to use rifampicin resistance as a surrogate marker for MDR TB in the study area. The prevalence of DRTB in Ethiopia has shown that the current WHO estimates for new and previously treated TB patients were low, and this proves that the burden of DR TB in Ethiopia could probably be bigger than what is anticipated and might jeopardize the success of the national TB control program. The possible justification for such widely varied differences might be due to differences in study design and setting, diagnostic tools employed, the variation introduced through time in the study, and differences in managing TB control program practices in different countries as well as sub-regions within the same nations. Further, probably all countries in recent times encountered policy changes in their national TB guideline to maximize the use of rapid TB diagnostics for effective triage and screening of active TB cases in the community which was initially limited to priority subpopulations.

Rifampicin resistance was determined in GeneXpert MTB/RIF Assay System on rpoB gene mutations in the 81 bp rpoB core region which are five overlapping regions labeled as probe A [codons 507–511], probe B [codons 511–518], probe C [codons 518–523], probe D [codons 523–529] and probe E [codons 529–533] that are collectively complementary to the entire RRDR. It was claimed for so long that, mutations conferring Rifampicin resistance in the 81 bp RRDR of the rpoB gene contribute to over 96% of *M. tuberculosis* and it is usually considered a mutation hot spot region [22, 45]. In our study, however, RR TB detected in the RRDR is 87.5%. Whilst, 12.5% of RR TB evolve to escape and was detected outside of the RRDR. Further, in this study, the commonest proportion of [62.5%] mutation conferring RR TB was located in probe E which is covering from 529–533 codons. A similar finding was reported in the previous studies in Ethiopia by Alemu A, et al., et al., (81%, 2020) [46], in Nigeria by Ochang EA, et al., (60.3%%, 2016) [47], in Uganda by Mboowa G, et al., (58.3%, 2014) [48], in India by Kaur R, et al., (56.1%, 2016) [29] and by Reddy R, et al., (54.9%, 2017) [49], in Pakistan by Ullah I, et al., (76.9%, 2016) [50], in Bangladesh by Rahman A, et al., (64.8%, 2016) [51] and Uddin MKM, et al., (64.8%, 2020) [52], and in Zimbabwe by Metcalfe JZ, et al., (60.4%, 2016) [53]. A similar prominent mutation conferring RR in the probe E region was also reported in Malawi by Chikaonda T, et al., (35.9%, 2017) [54] but this was with an exceptionally and equally shared mutation within the probe B region.

On the other hand, in our study, the second prominent proportion of [25.0%] mutation conferring RR TB was identified in probe D which covers 523–529 codons. Such a similar finding was also observed in previous studies conducted in Ethiopia by Alemu A, et al., (10%%, 2020) [46], in India by Reddy R, et al., (18.1%, 2017) [49], in Nigeria by Ochang EA, et al., (17.2%%, 2016) [47], and in Malawi by Chikaonda T, et al., (23.4%, 2017) [54]. However, most of the previous studies conducted in African and Asian regions indicated that following probe E, the commonest mutation conferring RR TB was in probe B and then followed by probe D as demonstrated by Kaur R, et al., (20.7%, 2016) [29], Uddin MKM, et al., (15.1%, 2020) [52],

**Table 4. Comparison of this study with the prevalence of missed probes conferring RR TB using the GeneXpert MTB/RIF assay in 10 studies conducted in different countries.**

| Study and Country | Total Missed Probes | Mutations conferring RR TB in RRDR | | | | | | Detected Outside of RRDR |
| --- | --- | --- | --- | --- | --- | --- | --- | --- |
| | | Probe A | Probe B | Probe C | Probe D | Probe E | Mixed Missed Probe | |
| Alemu A, et al., 2020, Ethiopia | 100 | . | 3 | . | 10 | 81 | . | 6 |
| Kaur R, et al., 2016, India | 130 | 10 | 27 | 1 | 18 | 73 | 1 | 0 |
| Reddy R, et al., 2017, India | 171 | 14 | 26 | 1 | 31 | 94 | 5 | 0 |
| Uddin MKM, et al.,2020, Bangladesh | 205 | 6 | 31 | 5 | 25 | 133 | 5 | 0 |
| Rahman A, et al., 2016, Bangladesh | 91 | 2 | 14 | 3 | 13 | 59 | 0 | 0 |
| Ullah I, et al., 2016, Pakistan | 408 | 5 | 44 | 6 | 34 | 314 | 5 | 0 |
| Ochang EA, et al., 2016, Nigeria | 58 | 2 | 8 | 0 | 10 | 35 | 3 | 0 |
| Mboowa G, et al., 2014, Uganda | 12 | 1 | 3 | 0 | 1 | 7 | 0 | 0 |
| Metcalfe JZ, et al., 2016, Zimbabwe | 43 | 2 | 6 | 5 | 4 | 26 | 0 | 0 |
| Chikaonda T, et al., 2017, Malawi | 64 | 2 | 23 | 1 | 15 | 23 | 0 | 0 |
| This Study Finding, 2022, Ethiopia | 24 | 0 | 0 | 0 | 6 | 15 | 0 | 3 |

Rahman A, et al., (15.3%, 2016) [51], Ullah I, et al., (10.7%, 2016) [50], Mboowa G, et al., (25.0%, 2014) [48], Metcalfe JZ, et al., (13.9%, 2016) [53], and Chikaonda T, et al., (35.9%, 2017) [54].

In this study, there was no mutation conferring RR TB hybridized at probe A, probe B, and probe C, probably because this particular site of RRDR is less susceptible to mutations conferring RR TB or might be a less common mutation of these probes in the current study geographic area. Similarly, the absence of probe C mutation was reported in Nigeria by Ochang EA, et al., in 2016 [47] and in Uganda by Mboowa G, et al., in 2014 [48]. Further, mutations conferring RR TB in probe A and probe C were less commonly reported in studies done in different countries as reported by Kaur R, et al., in 2016 [29], Uddin MKM, et al., in 2020 [52], Rahman A, et al., in 2016 [51], Ullah I, et al., in 2016 [50], Mboowa G, et al., in 2014 [48], and Chikaonda T, et al., in 2017 [54] (Table 4).

Similar to what was previously reported by Alemu A, et al., (6.0%, 2020) [46], a significant proportion of [12.5%] mutations conferring RR TB were identified outside of RRDR. However, the existence of an estimated less than 5% rpoB gene mutations outside RRDR was reported in previous studies on RR TB isolates [22.24]. A plausible explanation could be the delta Ct (ΔCT) max and low DNA amount. This has happened in this study where the ΔCT max was ≥4. Moreover, mutations in the rpoB gene could be different in different regions of the world having varied TB endemicity. However, it needs further study. Further, probe A, probe B, and probe C were not hybridized in all cases which is an important observation and would mean that in this geographical region, this probe has no or minimal utility. Nevertheless, neither the predominance of Probe E nor the assay scope provides details of specific mutations in the rpoB gene, limiting its value as an epidemiological tool to study mutations conferring RR TB. However, it could be used to assess trends over time, identify pockets of transmission, and outbreak investigation especially when mutations conferring RR TB occur outside the probe E region.

In this study, the proportion of males with MTB infection was higher among the study subjects involved [55.1%]. A similar observation has been reported in previous studies in Ethiopia by Alemu A, et al., (55%, 2020) [46] and Wasihun AG, et al., (57.7%, 2020) [27] and in Indonesia by Christopher PM, et al., (62%, 2019) [44]. On the other hand, a slightly higher proportion of RR TB was detected in females in this study [54.2%]. A similar finding was reported from Ethiopia by Arega B, et al., in 2019 [55] and Araya S, et al., in 2020 [56]. Nevertheless, the

reasons for the increased risk of MDR TB among women were not entirely understandable and require further investigation. However, despite clear evidence of substantial sex disparities in the burden of TB, how these disparities extend to MDR or RR TB acquisition is not well understood. Arguably, in many studies conducted to date, the risk of MDR or RR TB development among those with TB is the same for males as for females [57].

The association between age and TB infection and acquisition of MDR/RR TB is not consistent all over the world, and our finding is also in line with this assertion. The most likely justification could be the nature of the population structure in different nations may account for the inconsistent conclusions. In this study history of previous TB treatment has been noted to be a major risk factor for the development of drug resistance and underscores the continual spread of DRTB in the community as there is a vivid gap between the emergence of MDRTB and RR TB cases and treatment enrolment in Ethiopia.

We acknowledge certain limitations in this study. Neither mycobacterial culture and phenotypic drug susceptibility, nor target sequencing of rpoB gene analysis were available at this time. Thus, we could not estimate the proportion of discordant analysis against reference testing to diagnose MDR TB/RR TB, as well as we could not establish the specific rpoB gene mutations to ensure the specificity and sensitivity of the assay to detect mutations in the rpoB core region.

## Conclusion

The proportion of RR-TB we observed in this study was high. Similarly, a higher proportion of RR TB was detected outside of the RRDR. Moreover, a significant number of the GeneXpert MTB/RIF Assay probes were identified as unhybridized and this critical observation would mean that most of the probes had no or minimal utility in this geographical region. This urges immediate action to reshape and institute acceptable TB triage and definite diagnostic algorithm in Ethiopia since early screening and diagnosis of cases with quality-assured, rapid, and molecular diagnostics solutions could substantially reduce the burden of TB diseases. This will also reduce the time to treatment initiation as well as to prevent transmission of RR TB in the community.

## Supporting information

**S1 File. Sociodemographic and GeneXpert assay result.**
(XLSX)

## Acknowledgments

The authors would like to acknowledge the Department of Microbial, Cellular and Molecular Biology, Faculty of Life Sciences, College of Natural and Computational Sciences, Addis Ababa University, St. Paul's Hospital Millennium Medical College and TB Reference Laboratory at Saint Peter TB Specialized Hospital for their arrangements and unreserved support during this research work. We are also grateful to acknowledge the study participants for their willingness and involvement in this study.

## Author Contributions

**Conceptualization:** Gizachew Taddesse Akalu, Belay Tessema, Beyene Petros.

**Data curation:** Gizachew Taddesse Akalu, Belay Tessema.

**Formal analysis:** Gizachew Taddesse Akalu.

**Investigation:** Gizachew Taddesse Akalu.

**Methodology:** Gizachew Taddesse Akalu, Belay Tessema, Beyene Petros.

**Project administration:** Gizachew Taddesse Akalu.

**Resources:** Gizachew Taddesse Akalu, Beyene Petros.

**Software:** Gizachew Taddesse Akalu.

**Supervision:** Belay Tessema, Beyene Petros.

**Validation:** Gizachew Taddesse Akalu, Belay Tessema, Beyene Petros.

**Visualization:** Gizachew Taddesse Akalu.

**Writing – original draft:** Gizachew Taddesse Akalu.

**Writing – review & editing:** Gizachew Taddesse Akalu, Belay Tessema, Beyene Petros.

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
