## [Decision Letter · Decision Letter 0]

6 Nov 2022

PONE-D-22-28840High Proportion of RR-TB and mutations conferring RR outside of the RRDR of the rpoB gene detected in GeneXpert MTB/RIF assay positive pulmonary tuberculosis cases, in Addis Ababa, EthiopiaPLOS ONE

Dear Dr. Akalu,

Thank you for submitting your manuscript to PLOS ONE. After careful consideration, we feel that it has merit but does not fully meet PLOS ONE’s publication criteria as it currently stands. Therefore, we invite you to submit a revised version of the manuscript that addresses the points raised during the review process.

Please submit your revised manuscript by Dec 21 2022 11:59PM.  If you will need significantly more time to complete your revisions, please reply to this message or contact the journal office at plosone@plos.org. Please include the following items when submitting your revised manuscript:A rebuttal letter that responds to each point raised by the academic editor and reviewer(s). You should upload this letter as a separate file labeled 'Response to Reviewers'.A marked-up copy of your manuscript that highlights changes made to the original version. You should upload this as a separate file labeled 'Revised Manuscript with Track Changes'.An unmarked version of your revised paper without tracked changes. You should upload this as a separate file labeled 'Manuscript'.If applicable, we recommend that you deposit your laboratory protocols in protocols.io to enhance the reproducibility of your results. Protocols.io assigns your protocol its own identifier (DOI) so that it can be cited independently in the future. For instructions see: https://journals.plos.org/plosone/s/submission-guidelines#loc-laboratory-protocols. Additionally, PLOS ONE offers an option for publishing peer-reviewed Lab Protocol articles, which describe protocols hosted on protocols.io. Read more information on sharing protocols at https://plos.org/protocols?utm_medium=editorial-email&utm_source=authorletters&utm_campaign=protocols.

We look forward to receiving your revised manuscript.

Kind regards,

Frederick Quinn

Academic Editor

PLOS ONE

Journal Requirements:

Reviewers' comments:

Reviewer's Responses to Questions

**Comments to the Author**

1. Is the manuscript technically sound, and do the data support the conclusions?

Reviewer #1: Yes

Reviewer #2: Partly

2. Has the statistical analysis been performed appropriately and rigorously? 

Reviewer #1: I Don't Know

Reviewer #2: Yes

3. Have the authors made all data underlying the findings in their manuscript fully available?

Reviewer #1: Yes

Reviewer #2: Yes

4. Is the manuscript presented in an intelligible fashion and written in standard English?

Reviewer #1: Yes

Reviewer #2: Yes

5. Review Comments to the Author

Reviewer #1: Introduction

Ethiopia is among the 30 high TB burden countries with an estimated annual TB incidence rate of 132 per 100 000. The number of people infected with TB is 151 000, 9 900 HIV positive TB incidence, 19 000 HIV negative TB mortality, 2 500 HIV positive TB mortality, 591 individuals diagnosed with DRTB, and 7 cases of laboratory-confirmed pre-XDR TB or XDR TB [9]. Include the years for which this data is referring to.

Ethical approval

Can authors include the ethical approval number and date when it was granted.

Inclusion and Exclusion Criteria

The inclusion criteria were clinical suspicion of new presumptive pulmonary TB strong enough to warrant a direct sputum investigation and a positive result confirmed by Gene Xpert MTM/RIF. Take note of the inconsistence in writing the name of the GeneXpert MTB/RIF assay throughout the manuscript.

GeneXpert MTB/RIF Assay

Briefly, sputum samples were subjected to processing by approximately mixing with 8ml of GeneXpert sample reagent (SR) containing NaOH [5 - 10%] and isopropyl alcohol [10 – 20%] into the sputum container in a ratio of 2:1 V.V. The mixture was vigorously mixed by shaking the tubes 20 times and incubated for 15 minutes at 20–30 °C. This step has been repeated in between the incubation time. Take note of the sentence tense used here

GeneXpert MTB/RIF Assay Performance Characteristics and Result Interpretation

Generally, results from GeneXpert MTB/RIF Assay were obtained and assessed with three result types namely, MTB Detected, MTB Not Detected, and Invalid/Error/No result. What of MTB detected with Rifampicin resistance (MTB/RR TB)?

Quality Assurance

The internal quality control of the GeneXpert MTB/RIF Assay System was validated using M. tuberculosis H37Rv, a non-RIF resistant and known RIF resistant strain stored in -20oC. Which strain is this and how often was this done ?

Ethical Considerations

Institutional Review Board approval has also been obtained from the Research and Ethics Review Committee of Saint Peter TB Specialized Hospital for the commencement of the study. Take note of the sentence tense used here

Sociodemographic Characteristics of Study Participants

Overall, a total of 216 participants who had enrolled in this study were considered persons with presumptive TB upon clinical suspicion and bacteriologically confirmed with GeneXpert MTB RIF Assay. Does Xpert detect presence of the bacteria or bacterial DNA?

Reviewer #2: Dear Author(s),

I find this piece of work very interesting and useful as I myself did a similar work in Nepal samples looking at the Rifampicin resistance profile. I have to admit that though we saw mutations outside the RRDR, we did not pursue it further, but we always thought that these profiles will be important for enhancing the DNA probes in future. So, it is an important piece of work.

These are my recommendations:

1. I would like to see more clarity on the study design especially pertaining to data related to Rifampicin resistance outside the RRDR.

2. I would like to see more clarity on Figure 1, Figure 2 and Table 3.

3. What was the mutation detected outside RRDR? How was it confirmed as missense mutation? How was it detected? Do we have any experimental data? I could not find any concrete evidence.

4. Do you have any data about any co-mutations on RRDR along with the RR mutation outside this region?

I would greatly appreciate your responses for these questions.

6. PLOS authors have the option to publish the peer review history of their article (what does this mean?). If published, this will include your full peer review and any attached files.

Reviewer #1: **Yes: **Gerald Mboowa

Reviewer #2: No

---

## [Author Response · Author response to Decision Letter 0]

3 Dec 2022

Dear Dr. Frederic Quinn

Academic Editor

PLOS ONE

PONE-D-22-28840

High proportion of RR-TB and mutations conferring RR outside of the RRDR of the rpoB gene detected in GeneXpert MTB/RIF assay positive pulmonary tuberculosis cases in Addis Ababa, Ethiopia

PLOS ONE

Dear Dr. Frederic Quinn;

Thank you for your comments and for inviting us to submit a revised version. 

Comments by Academic Editor:

1. Comment on Journal Requirements:

• Thank you again and comments related to PLOS ONE’s style requirements including file naming are substantially accommodated.

2. Comment on Data Availability: 

• Thank you, and comments are accommodated.

• Our data are not under any of the legal or ethical restrictions and we wanted to share the anonymized metadata set as supporting information, “S1 File-Sociodemographic and GeneXpert assay result Xls” to support the validation of this study. Accordingly, we kindly request your editorial office update the “Data Availability” statement we provided during the first submission.

3. Ethical Approval Statement:

• Thank you for your comment. The comment is well accommodated just by moving it to the material and methods section, with content details on IRB, and informed consent. 

4. We take note of your comment regarding references and we validated it. 

• So far, we do not cite any retracted papers.

All comments from reviewers are accommodated point by point and included in the ‘Response to Reviewers’ file. 

Kindest Regards,

Gizachew Taddesse Akalu

Corresponding Author

---

## [Decision Letter · Decision Letter 1]

13 Dec 2022

High Proportion of RR-TB and mutations conferring RR outside of the RRDR of the rpoB gene detected in GeneXpert MTB/RIF assay positive pulmonary tuberculosis cases, in Addis Ababa, Ethiopia

PONE-D-22-28840R1

Dear Dr. Akalu,

We’re pleased to inform you that your manuscript has been judged scientifically suitable for publication and will be formally accepted for publication once it meets all outstanding technical requirements.

Kind regards,

Frederick Quinn

Academic Editor

PLOS ONE

Additional Editor Comments (optional):

Reviewers' comments:

Reviewer's Responses to Questions

**Comments to the Author**

1. If the authors have adequately addressed your comments raised in a previous round of review and you feel that this manuscript is now acceptable for publication, you may indicate that here to bypass the “Comments to the Author” section, enter your conflict of interest statement in the “Confidential to Editor” section, and submit your "Accept" recommendation.

Reviewer #1: All comments have been addressed

2. Is the manuscript technically sound, and do the data support the conclusions?

Reviewer #1: Yes

3. Has the statistical analysis been performed appropriately and rigorously? 

Reviewer #1: Yes

4. Have the authors made all data underlying the findings in their manuscript fully available?

Reviewer #1: Yes

5. Is the manuscript presented in an intelligible fashion and written in standard English?

Reviewer #1: Yes

6. Review Comments to the Author

Reviewer #1: In the revised manuscript, the authors have satisfactorily addressed all comments that I raised in the original version.

7. PLOS authors have the option to publish the peer review history of their article (what does this mean?). If published, this will include your full peer review and any attached files.

Reviewer #1: **Yes: **Gerald Mboowa

---

## [Editor Report · Acceptance letter]

19 Dec 2022

PONE-D-22-28840R1 

High proportion of RR-TB and mutations conferring RR outside of the RRDR of the rpoB gene detected in GeneXpert MTB/RIF assay positive pulmonary tuberculosis cases, in Addis Ababa, Ethiopia 

Dear Dr. Akalu:

I'm pleased to inform you that your manuscript has been deemed suitable for publication in PLOS ONE. Congratulations! Your manuscript is now with our production department. 

Kind regards, 

on behalf of

Dr. Frederick Quinn 

Academic Editor

PLOS ONE